# Optimal Transport for Offline Imitation Learning

**Yicheng Luo**
University College London

**Zhengyao Jiang**
University College London

**Samuel Cohen**
University College London

**Edward Grefenstette**
University College London

**Marc Peter Deisenroth**
University College London

## Abstract

With the advent of large datasets, offline reinforcement learning is a promising framework for learning good decision-making policies without the need to interact with the real environment. However, offline RL requires the dataset to be reward-annotated, which presents practical challenges when reward engineering is difficult or when obtaining reward annotations is labor-intensive. In this paper, we introduce Optimal Transport Reward labeling (OTR), an algorithm that can assign rewards to offline trajectories, with a few high-quality demonstrations. OTR's key idea is to use optimal transport to compute an optimal alignment between an unlabeled trajectory in the dataset and an expert demonstration to obtain a similarity measure that can be interpreted as a reward, which can then be used by an offline RL algorithm to learn the policy. OTR is easy to implement and computationally efficient. On D4RL benchmarks, we show that OTR with a single demonstration can consistently match the performance of offline RL with ground-truth rewards.

## 1 Introduction

Offline Reinforcement Learning (ORL) [17, 18] has made significant progress recently, enabling learning policies from logged experience without any interaction with the environment. ORL is relevant when online data collection can be expensive or slow. ORL algorithms can learn an improved policy that goes beyond the behavior policy that generated the data. However, ORL requires the existence of a reward function for labeling the logged experience, making direct applications of ORL methods impractical for applications where rewards are hard to specify with hand-crafted rules or when generating rewards for the dataset is potentially expensive. Therefore, enabling ORL to leverage unlabeled data is an open question with significant practical value. Several works have attempted to address this challenge. For example, Zolna et al. [24] proposes ORIL which learns a reward function that can be used to add reward labels to offline datasets, allowing unlabeled datasets to be used by offline RL algorithms.

Instead of having a reward function, providing expert demonstrations is more natural for practitioners compared to specifying a reward function. In robotics, providing expert demonstrations is fairly common, and in the absence of natural reward functions, 'learning from demonstration' has been used for decades to find good policies for robotic systems; see, e.g., [2, 1, 4, 9]. One such framework for learning policies from demonstrations is imitation learning (IL). Imitation Learning aims at learning policies that imitate the behavior given expert demonstrations. Behavior Cloning (BC) [22] is an IL approach that aims to recover the demonstrator's behavior directly by setting up an offline supervised learning problem. If demonstrations are of high quality and actions of the demonstrations are recorded, BC can work very well as demonstrated by Pomerleau [22], but generalization to new situations typically does not work well. Inverse Reinforcement Learning (IRL) [19] is another IL approach that learns an intermediate reward function that aims to capture the demonstrator's intent. State-of-the-art IRL (e.g., GAIL [12], DAC [15], PWIL [8]) methods can learn competent policies

with a small number of expert demonstrations. However, these algorithms focus on the online IL setting and are not applicable for learning from pure offline datasets.

In this paper, we introduce Optimal Transport Reward labeling (OTR), an algorithm that uses optimal transport theory to automatically assign reward labels to 'unlabeled' trajectories in an offline dataset, given one or more expert demonstrations. This reward-annotated dataset can then be used by offline RL algorithms to find good policies that imitate demonstrated behavior. OTR uses optimal transport to find optimal alignments between unlabeled trajectories in the dataset and expert demonstrations. The similarity measure between a state in the unlabeled trajectory and that of an expert trajectory is then treated as a reward label. These rewards can be used by any offline RL algorithm for learning policies from a small number of expert demonstrations and a large unlabeled dataset. We show that OTR consistently outperforms previous offline IL and reward learning approaches and can recover the performance of offline RL methods with ground-truth rewards with only a single demonstration on the D4RL [10] benchmark.

## 2   Offline Imitation Learning with Optimal Transport

**Problem Statement**   We consider learning in an episodic, finite-horizon Markov Decision Process $(\mathcal{S}, \mathcal{A}, p, r, \gamma, p_0, T)$ where $\mathcal{S}$ is the state space, $\mathcal{A}$ is the action space, $p$ is the transition function, $r$ is the reward function, $\gamma$ is the discount factor, $p_0$ is the initial state distribution and $T$ is the episode horizon. A policy $\pi$ is a function from state to a distribution over actions. The goal of reinforcement learning is to find policies that maximize episodic return. Running a policy $\pi$ in the MDP generates a state-action episode/trajectory $(s_1, a_1, s_2, a_2, \ldots, s_T) =: \tau$. We are interested in the problem of imitation learning from offline datasets. In this case, we assume that we have access to a small dataset of *expert* demonstrations $\mathcal{D}^e = \{\tau_e^{(n)}\}_{n=1}^N$ generated by an expert policy $\pi_e$ and a large dataset of *unlabeled* trajectories $\mathcal{D}^u = \{\tau_\beta^{(m)}\}_{m=1}^M$ generated by an arbitrary behavior policy $\pi_\beta$. We are interested in learning an offline policy $\pi$ combining information from the expert demonstrations and unlabeled experience, without any interaction with the environment.

**Reward Labeling via Wasserstein Distance**   Optimal Transport (OT) [6, 21] is a principled approach for comparing probability measures. The (squared) Wasserstein distance between two discrete measures $\mu_x = \frac{1}{T} \sum_{t=1}^T \delta_{x_t}$ and $\mu_y = \frac{1}{T'} \sum_{t=1}^{T'} \delta_{y_t}$ is

$$\mathcal{W}^2(\mu_x, \mu_y) = \min_{\mu \in M} \sum_{t=1}^T \sum_{t'=1}^{T'} c(x_t, y_{t'}) \mu_{t,t'}, \tag{1}$$

where $M = \{\mu \in \mathbb{R}^{T \times T'} : \mu \mathbf{1} = \frac{1}{T} \mathbf{1}, \mu^T \mathbf{1} = \frac{1}{T'} \mathbf{1}\}$ is the set of coupling matrices, $c$ is a cost function, and $\delta_x$ refers to the Dirac measure for $x$. The optimal coupling $\mu^*$ provides an alignment between the samples in $\mu_x$ and $\mu_y$. Unlike other divergence measures (e.g., KL-divergence), the Wasserstein distance is a metric and it incorporates the geometry of the space. Let $\hat{p}_e = \frac{1}{T'} \sum_{t=1}^{T'} \delta_{s_t^e}$ and $\hat{p}_\pi = \frac{1}{T} \sum_{t=1}^T \delta_{s_t^\pi}$ denote the empirical state distribution of an expert policy $\pi_e$ and behavior policy $\pi$ respectively. Then the (squared) Wasserstein distance

$$\mathcal{W}^2(\hat{p}_\pi, \hat{p}_e) = \min_{\mu \in M} \sum_{t=1}^T \sum_{t'=1}^{T'} c(s_t^\pi, s_{t'}^e) \mu_{t,t'} \tag{2}$$

can be used to measure the distance between expert policy and behavior policy. Let $\mu^*$ denote the optimal coupling for the optimization problem above, then eq. (2) provides a reward signal

$$r_{\mathrm{ot}}(s_t^\pi) = -\sum_{t'=1}^{T'} c(s_t^\pi, s_{t'}^e) \mu_{t,t'}^*, \tag{3}$$

which can be used for learning policy $\pi$ in an imitation learning setting.

**Imitation Learning Using Reward Labels From Optimal Transport**   We leverage the reward function from eq. (3) to annotate the unlabeled dataset with reward signals. Computing the optimal alignment between the expert demonstration with trajectories in the unlabeled dataset allows us

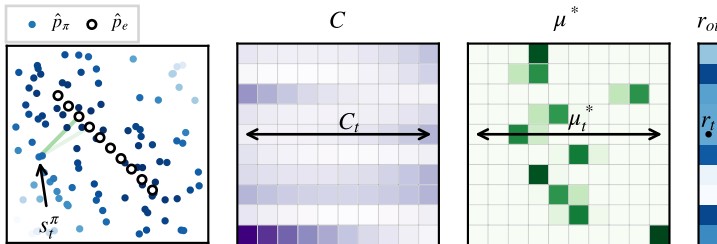

Figure 1: Illustration of the computations performed by OTR. In this example, we consider an MDP with a two-dimensional state-space ($|\mathcal{S}| = 2$). We have two empirical state distributions from an expert $\hat{p}_e$ with samples $\{s_{t'}^e\}_{t'=1}^{T'}$ ($\circ$) and policy $\hat{p}_\pi$ with samples $\{s_t^\pi\}_{t=1}^T$ ($\bullet$) as denoted by points in the leftmost figure. OTR assigns rewards $r_{ot}$ (blue) to each sample in the policy's empirical state distribution as follows: (i) Compute the pairwise cost matrix $C$ (purple) between expert trajectories and trajectories generated by behavior policy; (ii) Solve for the optimal coupling matrix $\mu$ (green) between $\hat{p}_\pi$ and $\hat{p}_e$; (iii) Compute the reward for $s_t^\pi$ as $r_{ot}(s_t^\pi) = -C_t^T\mu_t^*$. Consider for example a state $s_t^\pi \in \hat{p}_\pi$; the row $C_t$ in the cost matrix corresponds to the costs between $s_t^\pi$ and $\{s_{t'}^e\}_{t'=1}^{T'}$. $\mu_t^*$ represents the optimal coupling between $s_t^\pi$ and the expert samples. The optimal coupling moves most of the probability mass to $s_3^e$ and a small fraction of the mass to $s_4^e$ (green lines in the leftmost figure).

to assign a reward for each step in the unlabeled trajectory. Figure 1 illustrates the computation performed by OTR to annotate an unlabeled dataset with rewards using demonstrations from an expert. The pseudo-code for our approach is given in algorithm 1. In practice, we solve the entropy-regularized OT problem with Sinkhorn's algorithm [6, 7] to find the optimal coupling. Once we have computed the intrinsic rewards, we can use an offline RL algorithm that requires reward-annotated offline datasets. Unlike prior works that compute rewards using online samples [8, 5], we compute the rewards entirely offline, prior to running offline RL training, avoiding the need to modify any part of the downstream offline RL pipeline. This means that OTR can be combined with any offline RL algorithms, providing dense reward supervision required by the downstream algorithms.

**Advantages** Our approach enjoys several advantages. (1) our approach does not require training separate reward models or discriminators, which may incur higher runtime overhead. By not having to train a separate parametric model, we avoid hyper-parameter tuning on the discriminator network architectures. (2) Unlike other approaches, such as GAIL or DemoDICE, our approach does not require solving a minimax optimization problem, which can suffer from training instability [20]. (3) Our approach is agnostic to the offline RL methods for learning the policy since OTR computes reward signals independently of the offline RL algorithm.

## 3 Experiments

We demonstrate that OTR can be effectively combined with an offline RL algorithm to learn policies from a large dataset of unlabeled episodes and a small number of high-quality demonstrations. Since OTR is only a method for reward learning, it can be combined with any offline RL algorithm that requires reward-annotated data for offline learning. In this paper, we combine OTR with the Implicit $Q$-Learning (IQL) algorithm [16].

**Baselines** We compare OTR+IQL with the following baselines (1) *IQL (oracle)* [16]: the Implicit $Q$-learning algorithm using the ground-truth rewards provided by the D4RL datasets, (2) *DemoDICE* [14] an offline imitation learning algorithm based on distribution matching. (3) *ORIL* [24]: a reward function learning algorithm based on using positive-unlabeled (PU) learning, (4) *UDS* [23]: a simple baseline where the expert demonstrations have ground-truth rewards and the unlabeled dataset is relabeled with the minimum rewards from the environment. We use IQL as the same offline RL backbone (IQL) for both UDS and ORIL so that we can focus on comparing the performance difference due to different approaches used in generating reward labels.

**D4RL Locomotion** We evaluate the performance of OTR+IQL on the D4RL locomotion benchmark [10]. For the expert demonstrations, we choose the best episode from the D4RL dataset based on the episodic return. To obtain the unlabeled dataset, we discard the original reward information in the dataset. Table 1 compares the performance between OTR+IQL with the other baselines with only a single expert demonstration on the locomotion datasets in D4RL. Overall, OTR+IQL performs

best compared with the other baselines in terms of aggregate score over all of the datasets we used, recovering the performance of IQL with real rewards provided by the dataset. While we found that other baselines can perform well on some datasets, the performance is not consistent across the entire dataset and can deteriorate significantly on some datasets. In contrast, OTR+IQL is the only method that consistently performs well for all datasets of different compositions. We also find that OTR is faster compared to algorithms that learn additional neural networks as discriminators (DemoDICE [14]) or reward models (ORIL [24]) thanks to using a GPU-accelerated Optimal Transport solver [7].

| Dataset | IQL (oracle) | DemoDICE | IQL+ORIL | IQL+UDS | OTR+IQL (ours) |
|---|---|---|---|---|---|
| halfcheetah-medium-v2 | 47.4 ± 0.2 | 42.5 ± 1.7 | **49.0 ± 0.2** | 42.4 ± 0.3 | 43.3 ± 0.2 |
| hopper-medium-v2 | 66.2 ± 5.7 | 55.1 ± 3.3 | 47.0 ± 4.0 | 54.5 ± 3.0 | **78.7 ± 5.5** |
| walker2d-medium-v2 | 78.3 ± 8.7 | 73.4 ± 2.6 | 61.9 ± 6.6 | 68.9 ± 6.2 | **79.4 ± 1.4** |
| halfcheetah-medium-replay-v2 | 44.2 ± 1.2 | 38.1 ± 2.7 | **44.1 ± 0.6** | 37.9 ± 2.4 | 41.3 ± 0.6 |
| hopper-medium-replay-v2 | 94.7 ± 8.6 | 39.0 ± 15.4 | 82.4 ± 1.7 | 49.3 ± 22.7 | **84.8 ± 2.6** |
| walker2d-medium-replay-v2 | 73.8 ± 7.1 | 52.2 ± 13.1 | **76.3 ± 4.9** | 17.7 ± 9.6 | 66.0 ± 6.7 |
| halfcheetah-medium-expert-v2 | **86.7 ± 5.3** | 85.8 ± 5.7 | 87.5 ± 3.9 | 63.0 ± 5.7 | **89.6 ± 3.0** |
| hopper-medium-expert-v2 | 91.5 ± 14.3 | 92.3 ± 14.2 | 29.7 ± 22.2 | 53.9 ± 2.5 | **93.2 ± 20.6** |
| walker2d-medium-expert-v2 | 109.6 ± 1.0 | 106.9 ± 1.9 | 110.6 ± 0.6 | 107.5 ± 1.7 | 109.3 ± 0.8 |
| locomotion-v2-total | 692.4 | 585.3 | 588.5 | 494.9 | 685.5 |
| runtime | 20m | 100m* | 30m | 20m | 22m |

* The runtime is measured with the original PyTorch implementation.

Table 1: D4RL performance comparison between IQL with ground-truth rewards and OTR+IQL with a single expert demonstration ($K = 1$). We report mean $\pm$ standard deviation per task and aggregate performance and highlight near-optimal performance in **bold** and extreme negative outliers in **red**. OTR+IQL is the only algorithm that performs consistently well across all domains.

**Antmaze & Adroit** We additionally evaluate OTR+IQL on the `antmaze-v0` and `adroit-v0` datasets following the same evaluation protocol. OTR+IQL again recovers the performance of IQL with ground-truth rewards. This suggests that OTR can learn from datasets with diverse behavior and human demonstrations even without ground-truth reward annotation. See appendix A.5 for additional results.

**Qualitative comparison of the reward predictions** Figure 2a provides a qualitative comparison of the reward predicted by OTR, ORIL, and UDS. OTR's reward prediction more strongly correlates with the ground-truth rewards from the environment. We also evaluate OTR's reward prediction on more diverse datasets, such as those in antmaze. Figure 2b shows the demonstrations we used in `antmaze-medium-play-v0` and the top unlabeled trajectories. OTR correctly assigns more rewards to trajectories that are closer to the expert demonstrations.

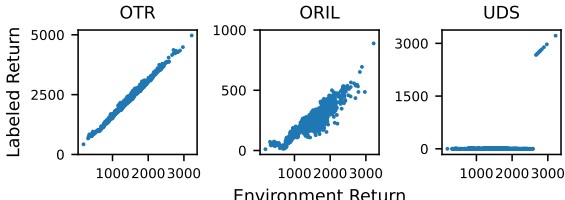 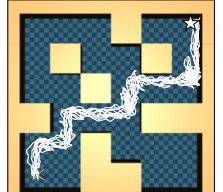 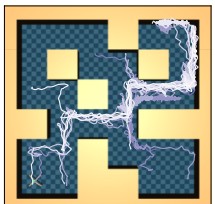

(a) Ground-truth return vs labeled return.          (b) Top trajectories selected by OTR.

Figure 2: Visualization of rewards predicted by OTR and baselines. (a) Qualitative differences between the rewards predicted by OTR, ORIL and UDS on `hopper-medium-v2`. (b) Visualization of top trajectories selected by OTR on `antmaze-medium-play-v0`. Left: Expert demonstrations. Right: ranking of trajectories according to rewards per step computed by OTR. Trajectories with lighter colors have higher rewards per step.

## 4 Conclusion

We introduced Optimal Transport Reward labeling (OTR), a method for adding reward labels to an offline dataset, given a small number of expert demonstrations. We demonstrate that OTR is computationally efficient and can recover the performance of offline RL with ground-truth rewards given only a single episode of expert demonstration.

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

# A Appendix

## A.1 Algorithm Description

The pseudo-code for OTR is presented in algorithm 1. OTR takes the unlabeled dataset $\mathcal{D}^u$ and expert demonstration $\mathcal{D}^e$ as input. For each unlabeled trajectory $\tau^{(m)} \in \mathcal{D}^u$, OTR solves the optimal transport problem for each, obtaining the cost matrix $C^{(m)}$ and optimal alignment $\mu^{*(m)}$ (line 3). OTR then computes the per-step reward label following eq. (3) (line 5). The reward-annotated trajectories are then combined, forming a reward-labeled dataset $\mathcal{D}^{\text{label}}$.

---

**Algorithm 1:** Pseudo-code for Optimal Transport Reward labeling (OTR)

    **Input:** unlabeled dataset $\mathcal{D}^u$, expert dataset $\mathcal{D}^e$
    **Output:** labeled dataset $\mathcal{D}^{\text{label}}$
**1** $\mathcal{D}^{\text{label}} \leftarrow \emptyset$
**2** **foreach** $\tau^{(m)}$ *in* $\mathcal{D}^u$ **do**                 // Label each episode in the unlabeled dataset
**3**      $C^{(m)}, \mu^{*(m)} \leftarrow \texttt{SolveOT}(\mathcal{D}^e, \tau^{(m)})$      // Compute the optimal alignment with eq. (2)
**4**      **for** $t = 1$ *to* $T$ **do**
**5**          $r_{\text{OT}}(s_t^{(m)}) \leftarrow -\sum_{t'=1}^{T'} C_{t,t'}^{(m)} \mu_{t,t'}^{*(m)}$      // Compute the per-step rewards with eq. (3)
**6**      **end**
**7**      $\mathcal{D}^{\text{label}} \leftarrow \mathcal{D}^{\text{label}} \cup (s_1^{(m)}, a_1^{(m)}, r_1^{\text{OT}}, \ldots, s_T^{(m)})$      // Append labeled episode
**8** **end**
**9** **return** $\mathcal{D}^{label}$

---

## A.2 Implementation

For OTR, we follow the recommendation by Cohen et al. [5] and use the cosine distance as the cost function. When there are more than one episode of expert demonstration, we perform a top-$K$ aggregation strategy with $K = 1$, i.e., we compute the optimal transport with respect to each episode independently and use the rewards from the expert trajectory that give the best episodic return. Similar to [8, 5], we squash the rewards computed by line 5 with an exponential function $s(r) = \alpha \exp(\beta r)$. This has the advantage of ensuring that the rewards consumed by the offline RL algorithm have an appropriate range since many offline RL algorithms can be sensitive to the scale of the reward values. Please refer to appendix A.3 for additional experimental details and hyperparameters.

We implement OTR in JAX [3]. For computing the optimal coupling, we use OTT-JAX [7], a library for optimal transport that includes a scalable and efficient implementation of the Sinkhorn algorithm that can leverage accelerators, such as GPU or TPU, for speeding up computations. Our IQL implementation is adapted from [16][1], and we set all hyperparameters to the ones recommended in the original paper. All of our algorithms and baselines are implemented in Acme [13].

An efficient implementation of OTR is made easy by JAX, which includes useful functionality that allows us to easily parallelize computations. Concretely, we JIT-compile the computation of rewards for one episode and further leverage the $\texttt{vmap}$ function to compute the optimal coupling between an unlabeled episode with all of the expert episodes in parallel. Efficiently parallelizing the computation of the optimal coupling requires that all the episodes share the same length. This is necessary both for parallelizing the computation across multiple expert demonstrations as well as for avoiding recompilation by XLA due to changes in the shape of the input arrays. To achieve high throughput for datasets with varying episodic length, we pad all observations to the maximum episode length allowed by the environment (which is 1000 for the OpenAI Gym environments) but set the weights of the observations to zero. Padding the episodes this way does not change the solution to the optimal coupling problem. Note that padding means that a 1M transition dataset may create more than 1000 episodes of experience, in this case, the runtime for our OTR implementation may be higher effectively due to having to process a larger number of padded episodes.

---

[1]$\texttt{https://github.com/ikostrikov/implicit\_q\_learning}$

As a result, it requires only about 1 minute to label a dataset with 1 million transitions (or 1000 episodes of length 1000)[2]. For larger-scale problems, OTR can be scaled up further by processing the episodes in the dataset in parallel. Our implementation of OTR and re-implementation of baselines are computationally efficient. Even so, the training time for IQL is about 20 minutes, so that OTR adds a relatively small amount of overhead for reward annotation to an existing offline RL algorithm.

### A.3 Hyperparameters

Table 2 lists the hyperparameters used by OTR and IQL on the locomotion datasets. For Antmaze and Adroit, unless otherwise specified by table 3 or table 4, the hyperparameters follows from those used in the locomotion datasets.

| | Hyperparameter | Value |
|---|---|---|
| | Discount | 0.99 |
| Network Architectures | Hidden layers | $(256, 256)$ |
| | Dropout | none |
| | Network initialization | orthogonal |
| IQL | Optimizer | Adam |
| | Policy learning rate | $3e^{-4}$, cosine decay to 0 |
| | Critic learning rate | $3e^{-4}$ |
| | Value learning rate | $3e^{-4}$ |
| | Target network update rate | $5e^{-3}$ |
| | Temperature | 3.0 |
| | Expectile | 0.7 |
| OTR | Episode length $T$ | 1000 |
| | Cost function | cosine |
| | Squashing function | $s(r) = 5.0 \cdot \exp(5.0 \cdot T \cdot r / |\mathcal{A}|)$ |

Table 2: OTR Hyperparameters for D4RL Locomotion.

| | Hyperparameter | Value |
|---|---|---|
| IQL | Temperature | 10.0 |
| | Expectile | 0.9 |
| OTR | Squashing function | $s(r) = 5.0 \cdot \exp(T \cdot r)$ |

Table 3: OTR Hyperparameters for D4RL Antmaze.

| | Hyperparameter | Value |
|---|---|---|
| Network Architectures | Dropout | 0.1 |
| IQL | Temperature | 0.5 |
| | Expectile | 0.7 |

Table 4: OTR Hyperparameters for D4RL Adroit.

The IQL hyperparameters are kept the same as those used in [16]. Note that IQL rescales the rewards in the dataset so that the same set of hyperparameters can be used for datasets of different qualities. Since OTR computes rewards offline, we also apply reward scaling as in IQL. For the locomotion datasets, the rewards are rescaled by $\frac{1000}{\text{max\_return}-\text{min\_return}}$ while for antmaze we subtract 2 to the rewards computed by OTR. The reward processing in antmaze is different from the one used by the original IQL paper (which subtracts 1) since the rewards computed by OTR have a different range.

The squashing function used by OTR is based on the one used in [8]. The antmaze squashing differs slightly from the one used in locomotion and adroit due to use of an earlier configuration. In practice, this should have minimal effect on the performance.

---

[2]Runtime measured on `halfcheetah-medium-v2` with an NVIDIA GeForce RTX 3080 GPU.

## A.4 Effect of the Number of Demonstrations

| locomotion-v2-total | $K = 1$ | $K = 10$ |
|---|---|---|
| DemoDICE | 585.3 | 589.3 |
| IQL+ORIL | 588.5 | 618.3 |
| IQL+UDS | 494.9 | 575.8 |
| OTR+IQL | **685.5** | **694.3** |
| IQL (oracle) | | 692.4 |

Table 5: Aggregate performances of different reward labeling algorithms with different numbers of expert demonstrations. OTR is the only algorithm that leads to an offline RL performance close to using ground-truth rewards.

We investigate if the performance of the baselines can be improved by increasing the number of expert demonstrations used. Table 5 compares the aggregate performance on the locomotion datasets between OTR and the baselines when we increase the number of demonstrations from $K = 1$ to $K = 10$. DemoDICE's performance improves little with the additional amount of expert demonstrates. While ORIL and UDS enjoy a relatively larger improvement, they are still unable to match the performance of IQL (oracle) or OTR in terms of aggregate performance despite using the same offline RL backbone. OTR's performance is close to IQL (oracle) even when $K = 1$ and matches the performance of IQL (oracle) with $K = 10$.

## A.5 Additional Experimental Results on Adroit and Antmaze

We evaluate OTR on additional datasets from the antmaze and adroit domains with varying number of expert demonstrations. The results are presented in table 6 and table 7. OTR consistently recovers the performance of IQL with ground-truth rewards on these datasets, largely independent of the number $K$ of expert demonstrations provided.

| Dataset | IQL (oracle) | $K = 1$ OTR+IQL | $K = 10$ OTR+IQL |
|---|---|---|---|
| door-cloned-v0 | 1.60 | 0.01±0.01 | 0.01±0.01 |
| door-human-v0 | 4.30 | 5.92±2.72 | 4.15±2.08 |
| hammer-cloned-v0 | 2.10 | 0.88±0.30 | 1.31±0.70 |
| hammer-human-v0 | 1.40 | 1.79±1.43 | 1.36±0.22 |
| pen-cloned-v0 | 37.30 | 46.87±20.85 | 42.68±24.98 |
| pen-human-v0 | 71.50 | 66.82±21.18 | 69.41±21.50 |
| relocate-cloned-v0 | -0.20 | -0.24±0.03 | -0.24±0.03 |
| relocate-human-v0 | 0.10 | 0.11±0.10 | 0.10±0.07 |
| adroit-v0-total | 118.1 | 122.16 | 118.78 |

Table 6: OTR+IQL Results on Adroit.

| Dataset | IQL (oracle) | OTR+IQL ($K = 1$) | OTR+IQL ($K = 10$) |
|---|---|---|---|
| antmaze-large-diverse-v0 | 47.5±9.5 | 45.5±6.2 | 50.7±6.9 |
| antmaze-large-play-v0 | 39.6±5.8 | 45.3±6.9 | 51.2±7.1 |
| antmaze-medium-diverse-v0 | 70.0±10.9 | 70.4±4.8 | 70.5±6.9 |
| antmaze-medium-play-v0 | 71.2±7.3 | 70.5±6.6 | 72.7±6.2 |
| antmaze-umaze-diverse-v0 | 62.2±13.8 | 68.9±13.6 | 64.4±18.2 |
| antmaze-umaze-v0 | 87.5±2.6 | 83.4±3.3 | 88.7±3.5 |
| antmaze-v0-total | 378.0 | 384.0 | 398.2 |

Table 7: OTR+IQL Results on Antmaze.

## A.6 Combining OTR with Different Offline RL Algorithms

In the main experiments, we evaluated OTR by pairing it with the IQL algorithm. In this section, we investigate if OTR can recover the performance of a different offline RL algorithm (TD3-BC) [11] using ground-truth rewards. We observe that (i) the performance from OTR+TD3-BC mostly matches those using the ground-truth rewards; (ii) the performance is fairly robust to the choice of the number of expert trajectories ($K = 1$ and $K = 10$ many expert demonstrations provide comparable performance). However, There are more variances on some datasets (e.g., `halfcheetah-medium-expert-v2`). Nevertheless, the differences are smaller compared to the baselines and OTR+TD3-BC still performs better than the baselines in terms of the aggregate performance.

| Dataset | TD3-BC (oracle) | OTR (K=1) | OTR (K=10) |
|---|---|---|---|
| halfcheetah-medium-expert-v2 | 93.5±2.0 | 74.8±20.1 | 71.6±23.1 |
| halfcheetah-medium-replay-v2 | 44.4±0.8 | 39.4±1.3 | 38.9±1.5 |
| halfcheetah-medium-v2 | 48.0±0.7 | 42.6±1.0 | 42.7±1.1 |
| hopper-medium-expert-v2 | 100.2±20.0 | 103.2±13.9 | 98.9±19.7 |
| hopper-medium-replay-v2 | 64.8±25.5 | 74.9±28.8 | 80.2±23.1 |
| hopper-medium-v2 | 60.7±12.5 | 66.4±10.3 | 69.8±13.9 |
| walker2d-medium-expert-v2 | 109.5±0.5 | 109.0±0.6 | 108.8±0.8 |
| walker2d-medium-replay-v2 | 87.4±8.4 | 69.7±16.4 | 67.4±20.6 |
| walker2d-medium-v2 | 83.7±5.3 | 76.9±5.4 | 78.0±2.6 |
| locomotion-v2-total | 692.3 | 656.9 | 656.4 |

Table 8: OTR+TD3-BC Results on MuJoCo.

## A.7 Importance of Using the Optimal Transport Plan

In the main experiments, we compute the rewards based on the optimal coupling computed by the Sinkhorn solver. The optimal transport plan is sparse and transports most of the probability masses to only a few expert samples. In this section, we investigate what happens if we use a suboptimal transport plan where each sample from the policy's trajectory is transported equally to each sample in the expert's trajectory. In this case, the reward function essentially boils down to computing the average costs with respect to all of the states in the expert's trajectory.

| Dataset | OTR+IQL | OTR+IQL (Uniform Plan) |
|---|---|---|
| halfcheetah-medium-v2 | 43.3±0.2 | 43.5±0.3 |
| hopper-medium-v2 | 78.7±5.5 | 80.5±2.3 |
| walker2d-medium-v2 | 79.4±1.4 | 77.6±1.5 |
| halfcheetah-medium-replay-v2 | 41.3±0.6 | 41.6±0.8 |
| hopper-medium-replay-v2 | 84.8±2.6 | 69.8±10.1 |
| walker2d-medium-replay-v2 | 66.0±6.7 | 62.2±14.4 |
| halfcheetah-medium-expert-v2 | 89.6±3.0 | 90.6±2.9 |
| hopper-medium-expert-v2 | 93.2±20.6 | 89.2±14.0 |
| walker2d-medium-expert-v2 | 109.3±0.8 | 106.0±5.9 |

Table 9: OTR with Uniform Transport Plan

Table 9 compares the performance of OTR+IQL using the optimal transport plan and uniform transport plan. We find that for many datasets, using the suboptimal uniform transport plan is sufficient for reaching good performance. This indicates that using a reward function based on the similarity of states from the policy and the expert can be a simple and effective method for reward labeling. However, note that the uniform transport plan can still underperform compared to using the optimal transport plan (e.g., `hopper-medium-replay-v2`). This shows that the optimal transport formulation enables better and more consistent performance.

