# OpenReview forum: "Optimal Transport for Offline Imitation Learning"
_NeurIPS.cc/2022/Workshop/Offline_RL — Offline RL Workshop NeurIPS 2022_

### Official Review · Reviewer_yGtR · 2022-10-17
**Review for Optimal Transport for Offline Imitation Learning**

**Rating:** 7
**Confidence:** 3

**Review:**

The paper presents a method for performing imitation learning by first computing rewards using a method called Optimal Transport Reward labeling (OTR) that computes the Wasserstein distance between an expert demonstration and an unlabeled one. The rewards are then used in combination with existing offline RL algorithms, for example, implicit Q-learning (IQL). The process is made computationally efficient through the use of GPU-accelerated optimal transport solvers.

Strengths:

- The method is conceptually simple, and the quantitative results are encouraging – OTR+IQL is able to recover the performance of IQL using oracle rewards for many of the D4RL tasks.
- It’s quite surprising and impressive that a single good expert demonstration can be used for the expert trajectory in this setup.
- The empirical evaluation compares to sensible baselines on the D4RL benchmark.
- The paper is clearly written and easy to understand!

Weaknesses:

- The introduction states that for behavior cloning, “generalization to new situations typically does not work well”, but I’m not sure I found the authors’ intuition in the paper for which aspect of OTR will improve generalization to new situations?
- It would be nice to see ablation type experiments showing that the optimal transport-based reward function performs better than simpler choices, but I understand that space is a constraint here.

While there are related works with a similar idea such as [1], this paper tackles a challenging and (as far as I am aware) novel situation of learning from a very small number of demonstrations, which I feel makes it of interest to this workshop.

[1]: Imitation Learning from Pixel Observations for Continuous Control (Cohen et. al)

---

### Official Review · Reviewer_Lte7 · 2022-10-20
**Interesting Paper**

**Rating:** 7
**Confidence:** 4

**Review:**

This paper proposes viewing differences in trajectory distributions as an optimal transport problem, hence relying on the Wasserstein distance to define the distance between an expert and a behavior policy. Given a distance measure, it can be used to define a reward for the trajectory.

I find such a view of the problem to be fairly interesting. The writing is clear and the experiments are nicely done. However, I was expecting the paper to go in detail about the connections between using the Wasserstein metric and other ways to arrive at a similar reward function. Since the reward is generated based on similarity in the behavior and expert policies, I wonder if other methods that estimate the transition dynamics would differ from the proposed method. Overall though, the paper motivates the problem well and does a good job in conveying the benefits of the method.